# Umandawa: Buddhist Transformation in Modern Sri Lanka

## Gihani De Silva

Department of Social Sciences, Faculty of Social Sciences and Languages, Sabaragamuwa University of Sri Lanka, Belihuloya P.O. Box 02, Sri Lanka; gihani.desilva@ssl.sab.ac.lk

**Abstract:** Charismatic Buddhist monks are instrumental in modernising Buddhism as they have been entrusted with an important role of resurrecting religion and Sinhala society throughout the course of Sri Lankan history. Ven. Pitaduwe Siridhamma, later known as Siri Samanthabhadra Arahat Thero, is known as a cosmopolitan modernist monk figure who envisions a modernised form of Buddhism in recent times, which is derived creatively from the discourses and practical ideals in traditional Buddhism. He went further by founding his style initiatives to address Buddhist transformations in modern Sri Lanka. Samanthabhadra revolutionised the monastery space, allowing his supporters to embrace ideals and incorporate them into their everyday life. His project includes a wide range of such activities, expanding opportunities for Buddhist women to pursue their religious vocations, favouring traditional forms of meditation over farming and similar activities. The mission to reform Buddhism in Sri Lanka is not entirely modern, as it also incorporates elements of tradition, as shown in the case study at Umandawa. The modernist ideals and societal demands that define contemporary Buddhism are reflected in the transformation of Buddhism in Sri Lanka.

**Keywords:** modern Buddhism; traditional Buddhism; Umandawa

## 1. Introduction

Heinz Bechert was one of the first scholars to use the term "modern Buddhism," which he divided into three stages: canonical Buddhism, traditional Buddhism, and contemporary Buddhism (Bechert 1973). In his proposal, Bechert's proposed new Buddhist trend was perfectly suited to the circumstances in late-nineteenth-century Ceylon, as demonstrated by his research. Bechert's ideas were later expanded upon by Gombrich and Obeyesekere (1988); Lopez (1995); and others (see McMahan 2008, pp. 7–8). The alterations to Sinhalese Buddhism were mainly a reaction to Christian missionary encroachment under British colonial rule on the island (see Bond 1988; Blackburn 2010, for Sri Lankan Buddhism in the 20th century). Western scientists and modern Buddhists both agreed that Buddhist philosophy was both logical and congruent with Western scientific principles (Teeuwen 2017, p. 1). There have been numerous movements and individuals involved in this massive transformation that has occurred across decades in Sri Lanka (Gombrich and Obeyesekere 1988). These idealised precolonial ideologies, such as *sāsanaya* (dispensation) and the *ārya* Sinhala myth (Sinhalese as an Indo-*āryan* ethnolinguistic group), facilitated Buddhism's adaptation to new realities due to colonialism's influence. Bechert points out that "Buddhist modernism" did not come without challenges, as it encountered opposition, not just from Christian missionaries, but also from "traditionalist Buddhists" in the early twentieth century. Bechert provides the example of "monastic landlordism," which was such a hindrance to the advancement of Buddhist modernism and made it impossible for the movement to accomplish the political aims of modernism. As a result of the reality of the institutions, new ideas would be relegated to the edges. As a result, modernists commonly appealed to the tradition, which meant returning to canonical Buddhism and appreciating historic Buddhist rites.

Following the experience of colonialism, Buddhism, like many other religions, was driven to situate itself on the periphery of state relations in an increasingly varied religious

landscape. By responding to "individualism, egalitarianism, liberalism, democratic values, and the impulse of social reforms," Buddhist modernism became much more complicated (McMahan 2008, p. 13). These Western ideas seemed to have fuelled mass meditation movements, laicisation, and so forth. However, some scholars have uncovered a substantial difference between early modernist Buddhists and later modernist Buddhists that they believe is noteworthy. While "earlier modernists sought to be modern by being scientific, philosophical, up-to-date, and patriotic," many late-twentieth-century and twenty-first-century modernists have instead emphasised creativity, spontaneity, mystery, and sensitivity to nature, as well as the interconnectedness of all life (McMahan 2008, p. 12).

In the case of the recent phase of Buddhist modernisation in Sri Lanka, there has been no mass movement, as there was in the late 19th and early 20th century Christian missionary and colonial impositions. Despite the fact that Buddhism was widely praised among other religions throughout the civil war (1983–2009) and in the immediate post-war years, its significance has been called into question with the exposure of the political ambitions of new Buddhist groups. To argue that Sinhalese Buddhism was not transformed during the post-war period would be an understatement. It seems to me that such innovative work is simpler to comprehend from the perspective of charismatic monks and the communities that surround them than it is in the case of institutionalised Buddhism.

The emergence of numerous charismatic monks is a new occurrence in Sri Lanka. In the 1970s, for example, Wickremeratne identified the dismal economic and political environment as having contributed to rapidly increasing unemployment (Wickremeratne 2006, pp. 162–63). During such uncertain times, many people were drawn to Ven. Ariyadhamma's Bodhi puja movement. Ven. Gangodawila Soma, on the other hand, revived the 21st century revivalist agenda (Frydenlund 2016, p. 107). This agenda includes a resurgence of the traditional Sinhalese Buddhist ethos. The traditional Sinhalese Buddhist ethos includes the following five precepts, with a special focus on condemnation of alcohol consumption, abandoning ritual superstition, and giving primacy to family (Berkwitz 2008). The Sinhala public was lured by a charismatic modern monk, Ven. Gandodawila Soma, an eloquent master of rhetoric, who utilised nearly every media to effectively portray his vision of Sri Lanka as an authentic Buddhist society (Holt 2019, p. 4). Soma and his supporters believed that Sri Lanka had always been a Buddhist country, and they wanted to see the Buddhist faith restored to its former glory.

Even if similar monks have emerged since then, it is doubtful that they have been able to make such social, political, and cultural changes in modern Sri Lanka. However, one of the characteristics of some of these latter monks is that they are attempting to remain out of politics. Ven. Kiribathgoda Gnanananda and Ven. Pitiduwe Siridhamma are two recent examples, who have followed very distinct routes. To promote his vision of the fundamental Buddhist discourse, Gnanananda, a modernist monk, utilises his Mahamevnawa media network to publicise his teaching (De Silva 2022). His popularity among young lay Buddhists in Sri Lanka and worldwide is soaring, as his teachings are widely focused on otherworldly practices due to his media savviness (Berkwitz 2016).

Certain monks adopted a nationalistic outlook by joining groups such as the JHU (Jathika Hela Urumaya), which strengthened Sinhala Buddhist nationalists within the ruling regime. However, these extremist monk figures and political parties have been condemned for their political agendas. According to Neena Mahadev, certain monastic figures were concerned about distancing themselves from politics and nationalism. As stated by Mahadev, Pitaduwe Siridhamma, " . . . The founder and the followers of the movement adamantly strive to distinguish their practices from nationalistic, exclusionary, and episodically violent drives that are often associated with Sinhala Buddhist revivalism" (Mahadev 2014, p. 129). However, my recent reading of these charismatic monks indicates that the new forms of Buddhism and social transformations in post-war Sri Lanka are reflected in their teachings. Assessing these projects illustrates the direction Sri Lankan Buddhists are currently taking in the aftermath of the civil war's atrocities against other ethnic groups. However, it is also fascinating that these monks retain their ties with militant

*bhikkhu*-s in certain aspects, as they are highly politicised and influential characters in the post-war Sri Lankan development of Buddhism.

Sri Lanka has no shortage of charismatic monks who have played important roles in rejuvenating the dispensation from time to time. While some of these monks embraced politics by consolidating the relationship between religion and state, others were primarily involved in social services. Ven. Pitaduwe Siridhamma, later known as Siri Samanthabadra Arahant Thero, took a distinct approach to envisaging his modernised version of Buddhism. He declared the state of nirvana the ultimate release from conditioned existence (*saṃsāra*), which was provocative in Sri Lanka, rather than giving in to nationalist concerns about the survival of Buddhism and protecting the nation's sovereignty. He furthered his claims with Umandawa after gaining popularity due to his dhamma discourses, which were seen as logical and scientific. I explore this project of Umandawa to learn about the Buddhist reforms he envisions.[1] He revolutionised the common monastery space by allowing people to embrace ideals by incorporating them into their everyday lives. His project includes a wide range of activities, from giving *vipassana* sermons to cultivating paddies. Their mission of transforming Buddhism in Sri Lanka is not entirely modern, as it also incorporates elements of traditional and conservative activities. This is because the transformation of Buddhism reflects modernist ideals and societal expectations as defining characteristics of contemporary Buddhism in Sri Lanka.

## 2. Ven. Pitiduwe Siridhamma to Siri Samanthabhadra Arahant Thero

One of the charismatic monks, Ven. Pitiduwe Siridhamma was born on 24 February 1975 in Pitiduwa, Galle, in the southern province of Sri Lanka, to Mr. and Mrs. Henry. He attended several schools and completed his advanced level exam at Richmond College, Galle, before enrolling at the University of Kelaniya. According to some reports, he held a Bachelor of Science in Microbiology (special) (Siri Samanthabhadra 2020). His university education turned him to the discourse of scientific Buddhism. He became a monk in 2000, despite his family's resistance, after developing a yearning to seek the ultimate truth in life. He immediately gained popularity as a Buddhist monk due to his attractive scientific and rational sermons presented via media. His incisive, analytical explanation of the dhamma for the average Buddhist used simple language. Moreover, his orientation toward scientific Buddhism seemed logical to the average Buddhist because it was seen as a response to social, political, and religious crises in contemporary society.

Ven. Siridhamma reappeared in a new garb after a period of silence during which his fame had grown. He may have visited various religious premises worldwide and met engaged Buddhists in this period. He began openly declaring and styling himself as an *arahant*. An *arahant* is one who has destroyed or abandoned four things: birth (*jāti*), the *āsavas*, the fetters (*saṃyojana*), and the burden (*bhāra*), has lived a holy life (*brahmacariya*), and achieved supreme knowledge (*abhiññāñāña*) (Engelmajer 2003, p. 34). In other words, the *arahant* is one who has comprehended the Buddha's Four Noble Truths. Declaring oneself as an *arahant* is hugely controversial in Sri Lanka, as well as in other Theravada Buddhist countries, and many people are sceptical of his claims of being an *arahant*.

According to Kapila Abhayavansha and Shyamon Jayasinghe, Samanthabhadra's self-proclaimed nirvana status raises methodological questions. As a result, he lost the respect of his monastic peers, and his ideas are routinely criticised by monastics and laity. More charges were levelled against him since he tried to act like a Buddha. As described by Abhayavansha, "that often he emerges in the style of the Buddha's appearance as described in the Sutta texts, resonating with the sermons of the Buddha at Jethawanaramaya" (Jayasinghe 2019). During my ethnographic observations at Umandawa monastery in October 2019, I observed the *katina* (robe offering) ritual and sermon delivered by Samanthabhadra. He walked in on a wreath of brightly coloured flower petals as he entered the newly constructed preaching hall. The *bhikkhu*-s and *bhikkhunī-s* who lived in the dwelling and laity collaborated for hours to produce this complex patterned piece of art. The laity sang a specially composed stanza in his honour upon his arrival at the assembly. The newly

erected Samanthabhadra statue was inaugurated on this occasion in the preaching hall. The golden painted statue of *arahant* Samanthabhadra was described as a gift offered by a couple of devout lay benefactors rather than being constructed following his directions. I witnessed a large number of people venerating the statue (for the *katina* ceremony at Umandawa see Figures 1–3). That statue, however, can be misidentified as a Buddha statue rather than his own. Samanthabhadra sat in front of this statue in a sermon, which resembled a royal throne, and which frequently appeared in his sermons (see Figures 3 and 4). Based on his behaviour, some believe that "Ven. Samanthabhadra constantly shows a deep preoccupation with himself (egoism), in fact, egotism, the notion that he is superior to all others" (Wasala 2017).

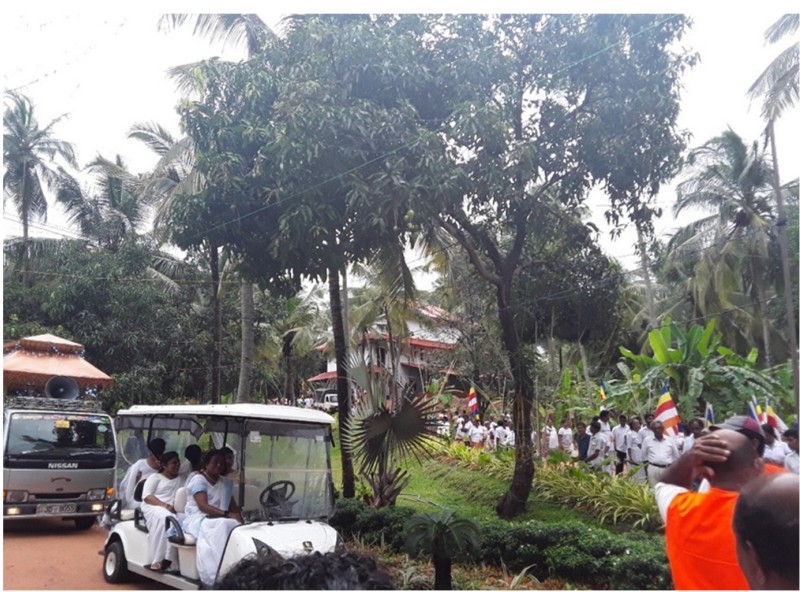

**Figure 1.** People prepare for the *Katina* procession. Photo by author. 19 October 2019, Umandawa Monastery, Kurunegala.

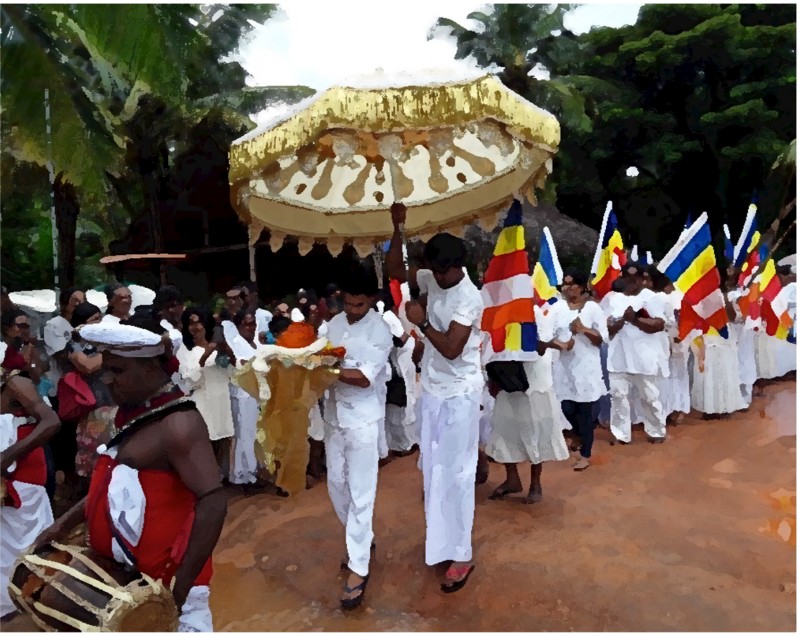

**Figure 2.** Katina robe is carried by the one of the primary donors in the procession under the pearl-encrusted umbrella. Photo by author. 19 October 2019, Umandawa Monastery, Kurunegala.

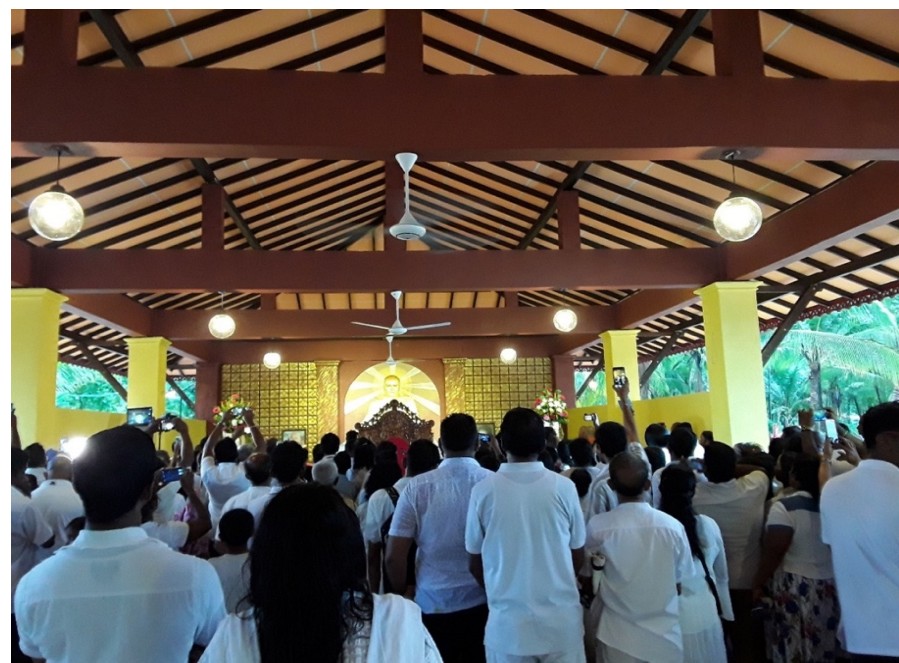

**Figure 3.** People gathered for the dhamma sermon in the preaching hall in front of the Samanthabhadra's statue. Photo by author. 19 October 2019, Umandawa Monastery, Kurunegala.

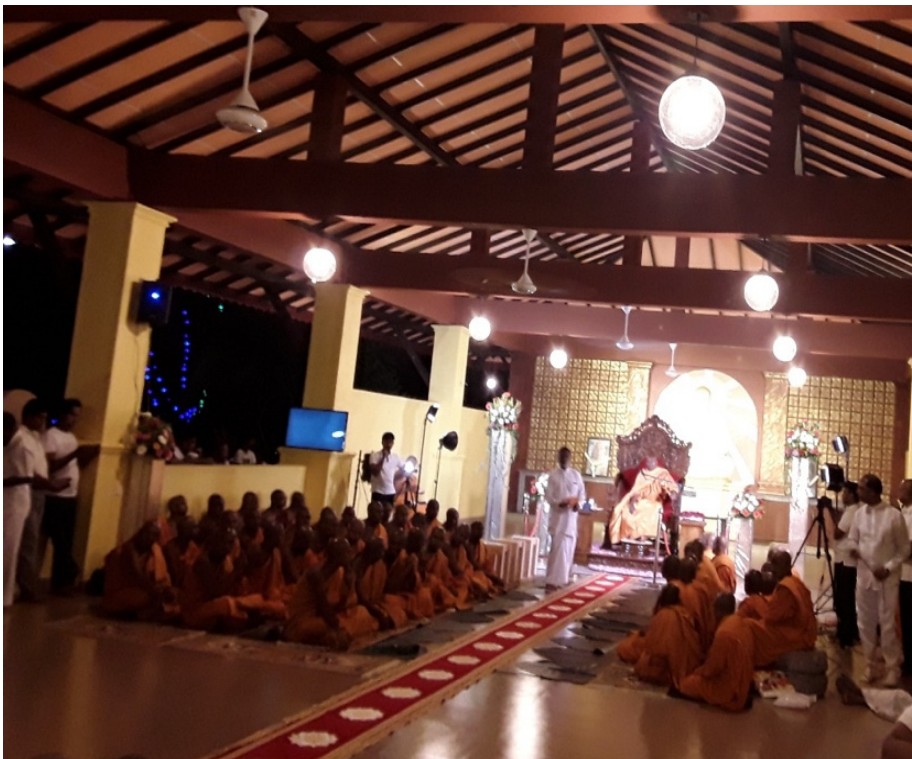

**Figure 4.** Samanthabhadra and his monastic disciples at the *katina* offering ceremony. Photo by author. 19 October 2019, Umandawa Monastery, Kurunegala.

It was stated by Jayasinghe that "There is not any doubt that the monk is a public performer" (Jayasinghe 2019). As recounted here, this attitude was evident in his performance at the dhamma sermon, as mentioned above. He was known for "his impressive articulation of ideas. The gift of the tongue is what carries him" (Jayasinghe 2019). Samanthabhadra is highly inclusive in his explication of the dhamma, speaking not only to Buddhists but also

to members of other religious and ethnic groups. "Many individuals who have arrived here have spent time in meditation facilities throughout Sri Lanka. They have listened intently to the sermons of some of the most accomplished monks in the country" (fieldnotes, 19 October 2019). He began his dhamma sermon, as previously recounted. The disciples who came to listen to his sermon added to this impression. They were not the regular *upāsaka* (male householders) and *upāsikā* (female householders) from the rural area where the monastery is located. They were mostly upper-middle-class people from the cities in their thirties to forties. They had come a long distance to attend this *katina* ritual and listen to his sermon. I came across one Muslim girl dressed in white attire listening to the sermon. She had spent considerable time travelling in search of a *guru* or a dhamma teacher. She said for the past three and a half years she has been visiting *āśramaya* (monasteries) to hear Samanthabadra's dhamma sermons, which she finds inclusive and grounded in science. What matters in this instance aside from the critique, is what his followers and devotees think about him. According to my findings, once a person begins to listen to his dhamma talks, he or she would continue doing so, regardless of any outside criticism, which can be described as a type of addiction.

The portrait of Samanthabhadra, the cosmopolitan modernist monk, represents those who have been instrumental in creating and maintaining a distinctively modernist form of Buddhism (McMahan 2008, p. 41). As McMahan states, "impulse toward modernisation, innovation, and reform are to a great extent inspired by modernist ideologies of the west, he does not embrace western modernity" (McMahan 2008, p. 36). Similarly, Samanthabhadra holds many traditional beliefs and practices, and seeks to adapt them to the modern world. He also advocates simplifying liturgy and making it more accessible to common people (Mahadev 2014). That is, Samanthabhadra continues to be rooted in his tradition while being explicitly inclined to modernised tendencies (McMahan 2008, p. 41) and indigenous modernity. He seems to have embraced many elements of western modernity but has not fully become assimilated. In other words, he has, in McMahan's words, "strategically adopt, reject, and transform elements of both modernity and tradition" (McMahan 2008, p. 42). He is aware that the majority of the people have not significantly transformed their ways of life, and this demonstrates that the country has reached a point where modernisation and traditionalism have found common ground after the open economy era of 1977.

## 3. Umandawa: The Global Buddhist Village

Although Samanthabadra represents a small minority of Buddhists, the decisive element in his success is due to "his visibility" (McMahan 2008, p. 41). His devotees swarm around him, and among them are celebrities, upper-middle-class wealthy individuals, and ordinary locals. Politicians with ambitions attend his sermons and other programs with the expectation of receiving his assistance. He will find a welcoming diaspora on the other side of the world (Sri Sadhaham Ashramaya 2022). What is pivotal to all these people is that they all benefit from Samanthabhadra's socioreligious efforts. I have seen upper-middle-class people, politicians, and celebrities (even controversial characters) receive special treatment at the Umandawa. When individuals are standing in line for meals, those persons are given priority, and their requirements are met by ignoring the others who were already standing in line at Umandawa (fieldnotes, 20 October 2019). Ordinary people are also welcome since the numbers of devotees are important, and donations, such as spectacles and packs of food, must be distributed.

Urbanites in Sri Lanka romanticise the idea of austere, forest-dwelling Buddhist monks, as opposed to domesticated urban monks, which is one of the trends that developed after the country's independence (Carrithers 1983). In urban areas of the country, people would regularly visit these ascetic monks to offer alms and strive to acquire merit. Samanthabhadra has drawn on these highly sensitive sentiments of city dwellers and provided them with a fresh perspective about domesticated monks living in the countryside. Although a true forest-dwelling monk cannot be found in his project, the resident monastics have

blended both the features of domesticated and forest-dwelling monks, forming a hybrid version.

The urban–rural divide that existed in Sri Lanka for decades is now diminishing. This is due to the transformation of certain parts of the urban society to rural areas, or rural areas becoming more urban. The population is not interested in absorbing the cultural features associated with the ruralisation of those areas but has romanticised rural living. In a way, the Umandawa resident monastics have filled a psychological space desired by these urban dwellers. That is, Umandawa speaks to all areas of current life in Sri Lanka (Jayasinghe 2019). Status and individual identity are increasingly defined not in terms of personal identity or social position, but rather in terms of lifestyle and possessions. Thus, Umandawa promotes a certain lifestyle that embraces the two forces of traditionalization and modernisation that have been converging in the recent past.

This Umandawa Maha Viharaya Monastery, also known as the Umandawa Global Buddhist Village, was founded in 2015. Umandawa means "the land of wise," originally derived from the Sanskrit word *ummaga*. The monastery is located in Malseripura, Kurunegala, spreading across 70 acres. The monastery was originally on abandoned land that had become overgrown with tropical jungle, but it has been transformed into an ecofriendly "walking monastery."

## 4. *Bhikkhunī*-s at Umandawa

The *bhikkhunī* order in Sri Lanka was founded in the third century BCE and prospered for a number of centuries, but then began to decline around the eleventh century CE. Until the late 1990s, there had been no attempt to resurrect the extinct order of *bhikkhunī*-s. However, colonial encounters with Sri Lankan Buddhism paved the way for the re-establishment of the defunct order of female renunciation, even though such initiatives had failed in the late 19th century (Bartholomeusz 1994, pp. 59–65). Tessa Bartholomeusz claims that Buddhist revivalism, opposition to British colonialism, and Christian missionary impositions were all factors that contributed to the emergence of the lay nuns' movement during this period. Influential lay elites such as Anagarika Dharmapala, who was instrumental in forming the Maha Bodhi Society, as well as theosophists such as Countess Miranda de Souza Canavarro favoured the resurgence of a female renunciant order in the country. Dharmapala wished to revive a certain type of *upāsikā* (women who observe moral precepts), who relinquish all of their family obligations (Bartholomeusz 1994, p. 55). Despite the efforts of Dharmapala and his supporters, these lay nuns and their projects remained on the margins of Buddhism. Later, many others took the initiative to establish an alternative female renunciant order titled "precept mothers", or *silmātā*-s (Bloss 1987).

Many individuals involved in the international movement revived the *bhikkhunī* order in Sri Lanka in the late 20th century, as alternative forms of female renunciation gained appeal in the country (Mrozik 2009). Sri Lankan Buddhist women have received higher ordination since across international and local monastic boundaries, and the island now possesses over 4000 *bhikkhunī*-s (Pathirana 2019). However, this newly established *bhikkhunī* order is not acknowledged by the government or monastic authority in Sri Lanka (Pathirana 2019). The Sri Lankan *bhikkhunī* community, on the other hand, is renowned worldwide since it was the first to be revitalized in Theravada Buddhism. This may be one of the motives for certain monks, especially famous ones, to support granting full ordination to women.

Women's ordination has gained prominence in Sri Lanka's contemporary mainstream Buddhism. Some liberal Buddhist monks have advocated the re-establishment of the *bhikkhunī* order due to its increased national and international recognition. Intriguingly, some of these monks were members of the Siyam *nikāya* (the leading monastic fraternity in the country). However, the chief monks (*Mahānāyaka*-s) of this fraternity have always been opposed to resurrecting the defunct *bhikkhunī* order in Sri Lanka. Individual monks from the Malwatta and Asgiriya fraternities (Siyam *nikāya*) assumed public leadership to

advocate women's ordination. This shows that the shift began with the conservative core, opposing the country's resuscitation of the *bhikkhunī* order.

According to McMahan, women play far more involved and visible roles in modern Buddhism than in earlier forms of the tradition, and modern Buddhism is also associated with the professional middle classes (McMahan 2008, p. 8). Samanthabadra, like many modernist monks, has supported the re-establishment of the *bhikkhunī* order, and he "favours reforms that provide more opportunities for women's religious vocation, and he supports greater lay involvement with the sangha, promoting new types of Buddhism . . . " (McMahan 2008, p. 35). His followers include many female disciples, most of whom are professional urban middle-class women. The members of this organization, much like the members of many other recently formed Buddhist groups, are circulating among the many Buddhist groups, switching allegiances frequently. In other words, adherents of a specific Buddhist group may visit a variety of dhamma masters and practitioners in search of everlasting happiness. It would appear that practitioners, teachers, and followers have a complete understanding of this reality, which entails followers drifting among various groupings. As described by George Bond, it is a prerequisite for the followers to believe that their *guru*-s, or teachers, have "experienced the advanced states of the Buddhist *marga* (Pāli: *magga*)," which is also true for Samanthabhadra's disciples (Bond 2003, p. 31). It seems to me that he decided to announce his nirvana because of these very followers. Declaring nirvana is a controversial act in a country such as Sri Lanka that follows the conservative Theravada tradition, because it implies that the monk claimant has approached the status equivalent to that attained by the Buddha.

The ordination of women at Umandawa under Samanthabadra's direction has been a very recent development. Some of these women kept asking him to ordain them as *bhikkhunī*-s. It is said that Samanthabadra was hesitant to ordain these women in the first instance, almost similar to the way the Buddha was hesitant to ordain his stepmother, Mahaprajapathi Gothami. Later, he proceeded to ordain women under his supervision. I believe Samanthabadra eventually decided to ordain women as part of his expansion of Umandawa, which currently consists of its fourfold communities (*bhikkhu, bhikkhunī, upāsaka,* and *upāsikā*).

The socioeconomic backgrounds of those women who sought ordination and ultimately received it under Samanthabadra are diverse. Most of them are from affluent backgrounds and are middle-aged, professional female lay followers who live in large cities. While some of them were able to persuade their families to support their choice, others had to divorce their husbands and often acted against the wishes of their families. These wealthy women have occasionally contributed their whole savings to help develop the nunnery grounds in Umandawa. Following *āśramavāsī bhikkhunī* Yashodhara's story, in brief, helps to explain some of these renunciants' motivations for becoming ordained under Samanthabadra.[2]

*Bhikkhunī* Yashodhara*'s* story reflects the characteristics of a truth-seeker, who met her spiritual goals despite various difficulties such as her family commitments, employment, or geographical location. She was a Middle Eastern migrant worker who worked at a Kuwaiti hospital. She earned all she desired but felt she was missing out on something in her life. She said her salvation was the Internet; she would be awake at night listening to the dhamma discourses on YouTube. As her quest progressed, she came across several "mud" postings (propaganda that defames someone) written against Samanthabadra, which piqued her interest in learning more about his dhamma. She had no clue at the time that this would be a turning point in her life, as she recounted in the following interview:

> I couldn't concentrate on my work after hearing his message. It was similar to drug addiction. My mind began to question, "Do you still want money after hearing so much [dhamma]?" I left behind all I had worked for over the years, including my gold jewellery. I didn't go home after arriving in Sri Lanka; instead, I went to *āśramaya*. The monastery had a Vipassana program. I was allowed to

speak. I opened my heart and felt a sense of comfort released from the burning. (Bhikkhunī Yashodhara, interview by author, 21 January 2020)

The defining trait of many of these new emerging Buddhist groups in modern Sri Lanka is that their members or followers seem to recognise a specific message conveyed by their *guru*-s. They unite behind these messages, which subsequently result in the groups gaining popularity. I noticed a similar trend in the Mahamevnawa *asapuwa* network, headed by Ven. Kiribathgoda Gnanananda, who is admired by his followers for a particular message (De Silva 2022). These new types of *guru*-s appear to have integrated ideas from various religions (including Christianity and views of Osho, etc.) to construct an impressive intellectual presence. For listeners, these messages could work almost like an addiction, similar to any other type of addiction, resulting in behavioural changes that make them avoid responsibilities, become obsessed with rules, detached from the real world, and show mood swings (Promises Behavioral Health/Behavioral Addictions 2013). Although it is not my intention to explain *bhikkhunī* Yashodhara's addictive behaviour, she remains addicted to the Internet, but this has been replaced by her religious addiction. She may have turned to religion to recover from the loneliness she suffered over the years. It seems she was not ready to settle down with her family, even though she had earned sufficient money. These migrant workers tend to continue working until their retirement. Thus, for Yashodhara, this religious addiction perhaps provided her with a good chance to escape from the suffering due to such continuous work. On the other hand, her family seemed powerless to stop her from becoming ordained since she may fulfil all the financial responsibilities to her family.

It appears that most of the renunciant women are middle-aged, while the path also attracts many young listeners. Some of the young women are celebrities, who are notorious for wearing fashionable dresses that reveal too much flesh, or clothes that are transparent in the temple, and the media has captured these women being photographed with Samanthabhadra, which draws more criticism. Whatever the nature of such criticism, neither Samanthabhadra nor his close disciples seem to be concerned, since the association with celebrities gives them and their group free publicity. On the other hand, it proves that even with the arrival of elaborately dressed women, their monastics are not affected and can act without any concern. This principle of detachment seems to apply in many ways at this temple, including their approaches to worldly luxuries or birthday celebrations (Colombo Telegraph 2014). According to their teachings, one should be able to endure luxurious worldly temptation while practising nonattachment.

In contrast to the above-mentioned type of worldly women who visit the temple and uplift their public profile, the renunciant women are a "closed minor group" (see Figures 4 and 5). The Umandawa *bhikkhunī*-s' story of ordination is somewhat fascinating in its positioning in the Umandawa project. The *sāmaṇērī* ordination of the Umandawa nuns was marked by a special event (Sujeewa 2021) in 2016, as it was conducted with the participation and guidance of one of the first Sri Lankan *bhikkhunī*-s, the late Ven. Bhikkhunī Kolonnawe Kusuma, who was ordained in 1996 in Sarnath, India (Dwyer 2018). Bhikkhunī Kusuma may have been asked to act as a preceptor because of her reputation as a well-known Buddhist scholar. However, according to some of my informants and social media posts, the relationship between preceptor and disciple (between Bhikkhunī Kusuma and the Umandawa nuns) did not continue after their ordination.

The Umandawa *bhikkhunī*-s' higher ordination, like their *sāmaṇērī* ordination, has distinct characteristics. To become fully ordained as *bhikkhunī*-s, the candidates collaborated with the Sakyadhitha Bhikkhunī training centre in Gorakana, which has links to the Naugala *bhikkhunī* community. They avoided the renowned Dambulla *bhikkhunī* community nearby, which is under the auspices of the monk Inamaluwe Sumangala. There may be a number of reasons why Umandawa *bhikkhunī*-s chose to associate with the Naugala community of *bhikkhunī*-s rather than with Dambulla's. Samanthabhadra has endorsed a caste-exclusive ordination for *bhikkhu*-s and *bhikkhunī*-s, just like what Ven. Inamaluwe Sumangala is publicly credited for with starting the caste exclusive ordination for *bhikkhu*-s and Sri Lankan *bhikkhunī*-s society by organising Buddhist women's higher ordination at his monastery.[3]

The reason may be because both monks are from the same Siyam *nikāya*, but are affiliated with separate fraternities within it. As both monks had challenged the Asgiriya and Malwatta caste-exclusive mother fraternities, they may have opted to pursue their *bhikkhunī* projects independently. Umandawa retained a small number of ordained *bhikkhunī*-s as his "close" group in his monastic community, compared to the more significant *bhikkhunī* communities such as those of Dambulla and Naugala.[4]

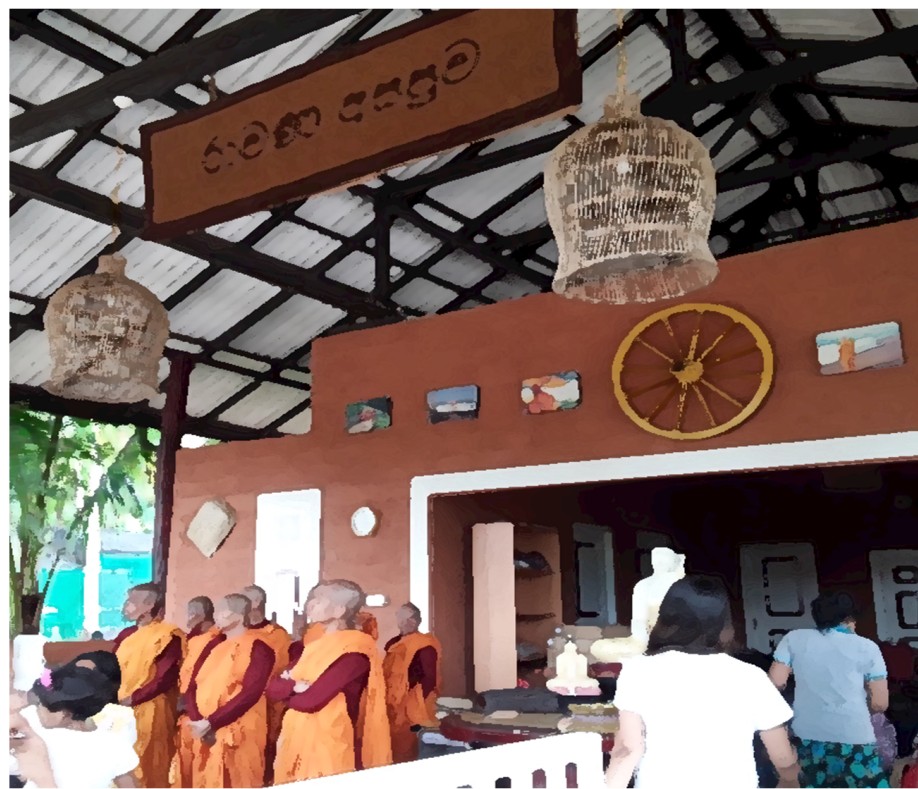

**Figure 5.** *Bhikkhunī*-s at Ravana *asapuwa.* Photo by author. 19 October 2019, Umandawa Monastery, Kurunegala.

The *bhikkhunī*-s in Umandawa live in a small, close-knit community and are directly supervised by Samanthabadra and his disciple monks (See Figure 5). Some of these monks claim that Umandawa *bhikkhunī*-s are quite distinct from the other Sri Lankan *bhikkhunī*-s, in that they remain under the tight supervision of monks in their community. The following summarises how a *bhikkhu* perceived the distinctiveness of this *bhikkhunī* community.

> There is no problem if the *bhikkhunī*-s' temple is next to a *bhikkhu* temple. However, due to some personality differences, there is an issue between men and women. In the vicinity of the Buddha, there was nothing like that. Because the *bhikkhunī*-s were living under the direction of the monks, they desired independence when they were by themselves. (Bhikkhu Dhammaveera, interview by author, 5 January 2020)

While charismatic monk leaders encourage Buddhist women to pursue higher ordination under their leadership and enable them to become ordained, monks are sceptical of potential conflicts between male and female monastics. For example, monks were concerned about personality issues, such as the likelihood of Buddhist nuns gaining dominance over the monks, which may potentially lead to a decline in the *sāsanaya* (dispensation). While the *bhikkhunī*-s in Sri Lanka are supported by *bhikkhu*-s to re-establish the *bhikkhunī* order in the country, *bhikkhunī*-s are obliged to live under certain conditions in the monastic community.[5] Failure to do so could jeopardise the *bhikkhunī*-s' relationship with the monks and in receiving their assistance. Ute Husken argues that strict adherence to the *Bhikkhunī*

Vinaya and ordination proceedings are essential for *bhikkhunī* communities in the Theravada tradition in the United States. She states that if they fail to follow these rules, they risk losing the support of the male *sangha* (Husken 2017, pp. 249–51).

Similarly, after the reintroduction of the *bhikkhunī* lineage, *bhikkhunī*-s in Sri Lanka have become subject to strict rules and are affiliated with the established monastic hierarchy. For instance, in the case of Umandawa *bhikkhunī*-s, the monks expect regular adherence to the Vinaya rules from their *bhikkhunī* counterparts. This starts from them following the *Aṣṭagarudharma*, which are the "heavy rules" that all *bhikkhunī*-s must follow, and as a result, even the most accomplished and senior *bhikkhunī* becomes subordinate to a monk who may be much junior to her. These heavy rules are one of the more controversial aspects in the history of female Buddhist renunciation (see Salgado 2008, pp. 182–83; Chiu and Heirman 2014, pp. 241–72). The following claim states that these strict rules have created the groundwork for women's higher ordination at Umandawa. One of my informants said, "Before they cut and shaved their hair, we gave them a piece of paper" (Bhikkhu, interview by author, 20 October 2019). This paper contains *Aṣṭagarudharma* and other regulations that a female candidate should follow before undergoing ordination. This remark describes an occurrence that Mahaprajapathi Gothami experienced in the face of her ambition to become ordained. Like many Buddhist women in the past who sought ordination, these present-day women joined the monastic community and accepted such requirements imposed by monks. Samanthabhadra has applied all these traditionally orthodox elements of Buddhist history to his recent endeavours. He seems to be most concerned with how he may represent the historical Buddha.

Since Buddhist nuns are subject to stringent regulations, as mentioned, it is intriguing to observe how those rules apply to their lives in Umandawa. Even today, the relationships between monastics are strictly governed by monastic rules. For example, Umandawa *bhikkhunī*-s (regardless of age) must rise when they meet a *bhikkhu*. According to one of the residing monks, he views this as a natural occurrence rather than an act of compulsion. He also stated that they do not require *bhikkhunī*-s to venerate a monk's feet every time they meet a monk for practical reasons. Instead, a *bhikkhunī* should converse with a monk respectfully by clasping her hands and bowing (*væňdagena katākaranavā*). One of the heavy rules states that *bhikkhunī*-s are placed under the direct authority of monks. However, while conducting research in Umandawa, I did not see the *bhikkhunī*-s bowing their heads to *bhikkhu*-s while conversing with them. The rule may be seen in this way as an idealised expectation of the *bhikkhu*-s towards the *bhikkhunī*-s. It is therefore understandable why Umandawa forbids its *bhikkhunī*-s from mixing with those of other *bhikkhunī* groups and tries to maintain conceptual autonomy over their notion of what a *bhikkhunī* should be.

Further research into the sermons of Samanthabhadra has revealed the origin of his stance on Buddhist women and *bhikkhunī*-s at Umandawa. For example, Samanthabhadra's disciples organised his dhamma sermon in Sweden and one of their questions there was why a *bhikkhunī* should honour a novice monk who had been ordained only for a day before the meeting with her. Samanthabhadra's response is below.

> Answer: Yes, if not [if Buddhist nuns don't revere monks] I wouldn't be able to remain serene [The monk and the crowd both laughed]. You know how badly behaved women are in such situations (*daňgalana dæňgalilla*). The same was held at the time of the Buddha. When given any power, women abuse it by distributing it to destroy everything in their way. That is the rationale behind the rules. Do not oppose them [laugh sarcastically]. I've discovered this from my personal experiences. Otherwise, this administration may be in a state of disarray.

> Our grandma has also told me a story. I once used a broom to hit a cat. Grandma warned me to stop hitting a male cat in the future [The monk shows how he struck the cat, and the audience chuckles]. Then I questioned why hitting a male cat was allowed, but hitting a female cat was not. She warned against approaching a male animal with a broom [Don't even think of raising your hand to hit a male animal]. It was a great honour [males]. That was the honour bestowed upon

males. That is how the *sāsanaya* should be. For the obvious fact that our society is patriarchal. A society dominated by males. This practice of male dominancy does not need to be changed. It's the way it is, and I have no intention of changing it. This tradition makes no difference what the notion is here [in Sweden], but it must exist. Alternatively, we may not be able to go and save our heads at all.

Question: Why aren't you open to this, considering your openness to so many other things?

Answer: It cannot be done in reality [with a loud laugh]. I don't always do things theoretically. I think about it in terms of practice. A woman's capacity for philosophy and spirituality is constrained. Simply said, that's how things stand. To accept it is necessary. It is feasible to become an arahant. That's OK. Men are far more capable than women in problem-solving, leadership, and management. These are things I've experienced first-hand. The monk should take the initiative in the *sāsana* tradition. (Buddha Rathnaya 2021)

This conversation is just one example of Samanthabhadra's attitudes toward women and Buddhist nuns. These attitudes are accepted at Umandawa in every aspect of its daily operation, as described in the following section. However, no one seems to be worried about this gender disparity, as members are busy concentrating on work at Umandawa. These attitudes are not spontaneous but come from Samanthabhadra's socioeconomic roots (he was born in a village), which were not replaced by his scientific dhamma teachings. Yet, somehow, he is very clever in using these existing gender practices to reaffirm them in his own initiatives.

## 5. Meditation into Everyday Practice

Many Buddhist practitioners and followers around the world consider meditation to be an important Buddhist activity. Recent scholarship has tracked the rise of lay meditation in South and Southeast Asia over the last century (Bond 2003; Braun 2013; Harris 2019; Jordt 2007). Nevertheless, meditation is also explained in the Pāli scriptures and is central to the Buddha's own experience and approach to achieving enlightenment.

As described by Michael Carrithers, there are renunciant monks who insist that the Buddha's entire teaching ought to be used as meditation instructions, with meditation in this sense meaning that it "'nourishes' 'develops' 'increases,' '*vaḍanavā*' the 'work' (*væḍa*)" (Carrithers 1983, pp. 222–23). In many settings, meditation has taken on a central role in Buddhist education. There are several benefits of meditation, such as "improving awareness, compassion, peace of mind, and even enhancing their practice of other faiths" (McMahan 2008, p. 184). Carrithers wrote, in 1983, that it was a largely prevalent idea in the 1950s in Sri Lanka that there was no mental capacity for people to attain nirvana within their present lifespan. However, the Galduva monks with whom he studied promoted the idea that "Buddhism still leads to nirvana," and what was thought to be an "unbridgeable abyss" at the time (in Carrithers' opinion) was being revived by these Galduva monks with their very serious attempts at a meditative life (Carrithers 1983, p. 222).

Meditation is a contemporary development and a key constituent of Buddhist modernism. Spiritual awakening is viewed as a long-term endeavour in Buddhist countries in Asia. A common idea held by Asian Buddhists is that we are living in an age of moral decline, and when advanced spiritual development is nearly impossible, more Buddhists are content with cultivating good *karma* (McMahan 2008, p. 40). Although meditation is considered vital for achieving the ultimate aim of the Buddhist path, it has been performed by only a small minority of monks and even fewer laypeople. That is, meditation has been traditionally seen as a demanding endeavour that should be undertaken only by highly trained persons who are willing to devote many years of their lives to its practice (McMahan 2008, p. 40).

Samanthabhadra, however, does not necessarily encourage meditation as a necessary practice for the path to liberation, despite the fact that many monastics and practitioners practise for the benefit of all humanity. Resident monks (or laity) in Umandawa do not practise any specific form of meditation, but Samanthabhadra directs them to attend his vipassana sermons instead. According to him and his monks, anyone who accumulates merit via meditation or rituals can ascend to the Brahma plane and then may descend into one of the four hells at some point, bringing about agony in an endless cycle of rebirth (Bhikkhu Dhammaveera, interview by author, 5 January 2020). Samanthabhadra also advises against carrying the dhamma on one's shoulders or causing needless tension over the dharma and its practice. Instead, they were being advised to focus on the task the person had been assigned with. This was explained by a resident monk as follows.

> Could you please tell me about someone who has meditated and gained nirvana? There is nothing like that. Everything stems from the mind. It is not necessary to meditate to control it. I've visited some meditation facilities where they even avoid young women who come to offer alms. When such monks see young women, they become annoyed. On the other hand, we are not bothered when women come and visit our *āśramaya*. That is our training. (Bhikkhu Dhammaveera, interview by author, 5 January 2020)

This young monk, who has attended a number of meditation centres (*araṇya*), is sceptical about monks who live and practice in them. Accordingly, instead of attaining detachment, ascetic monks who meditate suppress their *klesha* (afflictions), which is compared to pushing a rubber ball under water. Thus, they say it is pointless to repress such sensations by force; instead, a realisation of one's mind is required. As described in the following statement, they propose a training space where male and female monastics work and live together.

> I am the creator of my reality. In this world, I'm OK. I don't have somebody to fight with, but you can only examine your thinking when you are part of a group. We are told that we must work in groups with male monks. Many are young. That is the ideal opportunity to examine the mind. (Bhikkhuni Yashodhara, interview by author, 21 January 2020)

According to this *bhikkhunī*, the proximity of monks and nuns living together serves a necessary function. Umandawa encourages group activities, and each monastic member is assigned a certain responsibility. Their group projects are an excellent opportunity to examine one's cognitive process, which is akin to the practice of meditation as described in several faith traditions. What remains to be seen is how Samanthabhadra uses such modern psychological explanations to promote traditional labour at Umandawa.

For the Umandawa initiative to continue operating, a large amount of labour is required from both monastics and lay people. Regardless of the Vinaya rules, both female and male monastics are required to work whatever form of tasks Samanthabhadra assigns them to do. These activities include trimming trees, removing and burning weeds, cooking, cleaning, and chopping firewood. According to the resident monks, they ignore the Vinaya rules that might conflict with their labour activities in Umandawa. Furthermore, their resident monastics perform *poya* karma or *uposatha* observance [6] in their own chapter house without the assistance of outside monastics. Thus, for the monastic community of Umandawa, self-sufficiency and sustenance are more important than Vinaya rules and regulations.

Umandawa is always looking for volunteers since the monastic community is small and cannot sustain all its projects on its own. The professional supporters from the upper-middle class, who are primarily from urban areas, travel to Umandawa and willingly assist with the upkeep of the monastic property. They are given free meals by Umandawa, but they have to work all day to provide "*śrama dāna*" (free labour). The urban supporters listen to a vipassana program or regular dhamma discourses conducted by Samanthabhadra, during which some listeners are allowed to ask questions. These individuals may also

purchase organic food grown on the Umandawa premises and buy Samanthabhadra's books and CDs, earning income for the community.

Samanthabhadra undoubtedly infused the Umandawa project with a holistic, spiritual vision. A spiritual tourist package is part of his new endeavour, which is designed specifically for international tourists, rather than locals. Nevertheless, locals are given training in various agricultural and food-producing workshops. Male monks lead these international tourists, and the following is one of Umandawa's Facebook postings from when a Spanish group visited Umandawa in 2019.

> The day started with a local welcome drink commonly known as the 'shoe flower', known to be good for the heart, followed by a visit to *Seetha Wanaya*. They picked their own organically grown vegetables such as spinach, and among the other vegetables harvested were ladies' fingers and radish, which were cooked for lunch. They took immense pleasure to serve lunch to the monks. Subsequently, they took part in Sri Lankan cooking deliberations after enjoying a Sri Lankan local meal cooked from the organic produce of Umandawa. This was followed by Dhamma discussions with the head priest and other monks. The day ended with partaking in tea with local sweet specialities of *kokis* and *kævum*, which they were delighted to experience. They left Umandawa with an amazing experience of local traditions combined with exposure to the core Dhamma of Buddha. Their added experience was done with tracking and walks in a friendly environment. (Spiritual and Village Tourism—Umandawa 2019)

One of the primary goals of this program is to promote spiritual tourism, with a new effort focusing on local Sri Lankan cuisine. Spiritual tourists also visit Sri Lanka for pilgrimage, cultural heritage, meditation, and yoga. It is fascinating how the monastics are assigned tasks for this particular spiritual endeavour. While monks act as guides for international visitors, *bhikkhunī*-s prepare experimental, organic Sri Lankan food. Though these *bhikkhunī*-s are largely involved in preparing these numerous delicacies, Samanthabhadra always serves them to higher-level visitors based on his availability on the grounds.

Umandawa's popularity has soared in recent years due to measures taken by Samanthabhadra and his community. Despite his previous position of distancing himself from politics, as described by Mahadev, he publicly began to integrate politics and politician support into his initiative. For instance, Rajapakshas (Mahinda Rajapaksha and Namal Rajapaksha) were invited and involved in various efforts at Umandawa to gain publicity (Umandawa Global Buddhist Village 2022). As per his political connections, Samanthabhadra was able to utilize military men at Umandawa to assist him in his cultivation and other activities. Even though Umandawa received numerous compliments for his religious endeavour, he consistently received harsh criticism because of his attitude toward famous people. As a result, unlike other religious sites, Umandawa was assaulted by numerous individuals and a mob of village people claiming that the monastery was to blame for losing the support of the laity to local village temples (Sri Sadhaham Ashramaya 2022). However, the Umandawa community (monastics and laity) are working hard to make their initiatives a global phenomenon. Samanthabhadra is enthusiastic about this. It is rational and appealing in many ways. While transforming Sri Lanka's contemporary Buddhism, it embraces many conservative and traditional elements of Buddhism. The essence is that it offers something to everyone who enters the temple. It will never forget its essential qualities, nevertheless (See Figures 6 and 7).

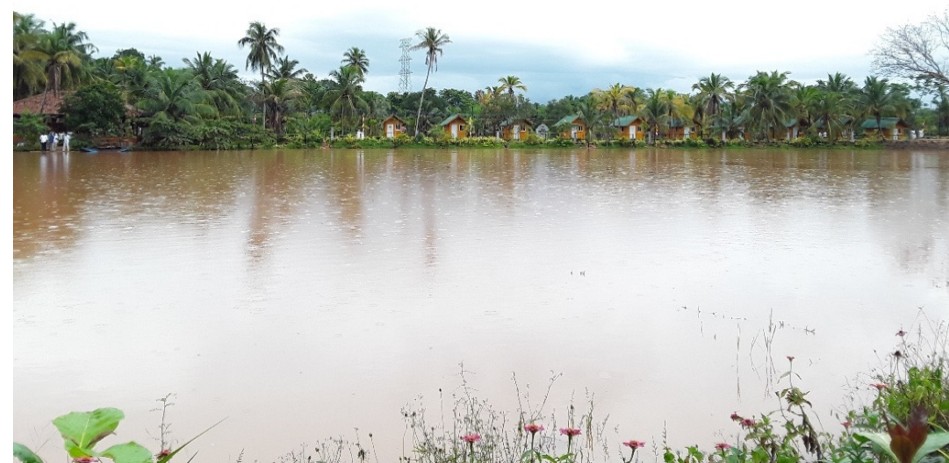

**Figure 6.** Lake flooded due to rain, Photo by author. 19 October 2019, Umandawa Monastery, Kurunegala.

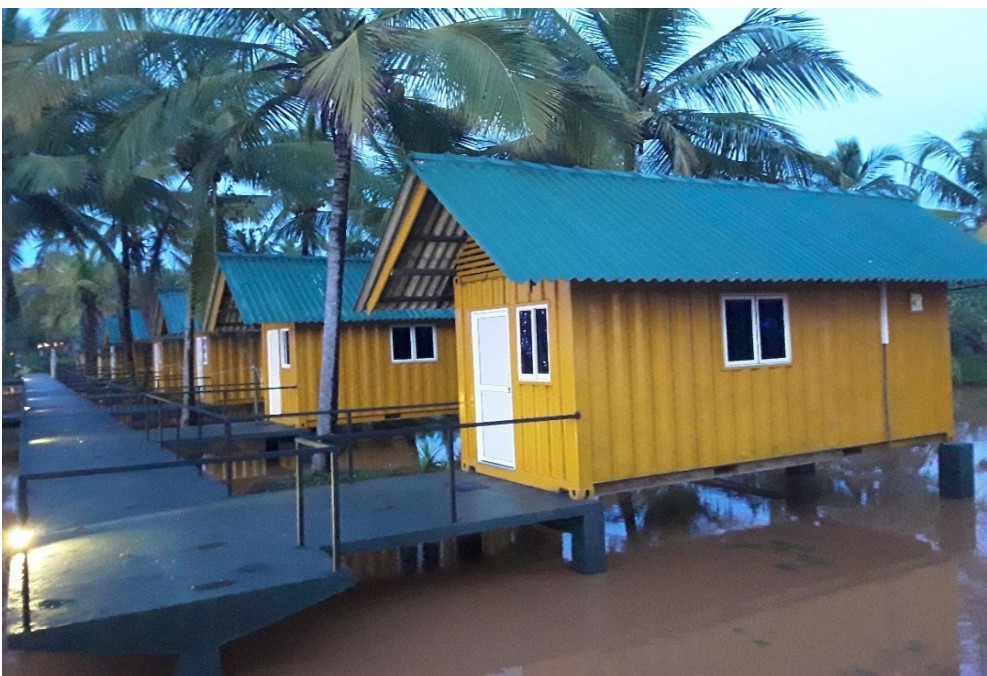

**Figure 7.** Newly built *kuti*-s (small huts or lodgings), Photo by author. 19 October 2019, Umandawa Monastery, Kurunegala.

### 6. Conclusions

This research paper examines a particular cosmopolitan monk figure, who is well-known in Sri Lanka's modernist discourse, and a model project he started. I intended to demonstrate how Ven. Samanthabhadra and his project initiative, Umandawa, represent one of the aspects of Buddhist transformation currently taking place in Sri Lanka. This undertaking (Umandawa) was approached from a scientific rationalism angle based on a western perspective and informed by his secular education. Paradoxically, it also exhibits the most conservative aspect of Theravada Buddhism, which is not seen in any other Buddhist monastery in Sri Lanka. The ordination of women at Umandawa was one of the crucial fundamental elements of the present discussion. Samanthabhadra favours expanding choices for Buddhist women to pursue their monastic vocation, including opting for higher ordination opportunities. While there is a possibility for *bhikkhunī*-s on the island

to evolve into an autonomous community of female renunciants via education and spiritual growth, the *bhikkhunī*-s living in Umandawa have limited horizons in their monastic career. Umandawa also encourages lay involvement with the sangha, reimagining the conventional communal activities in new areas such as agriculture. They do not promote the traditional meditation practice but require their lay adherents to concentrate on labour-intensive tasks in their agricultural initiatives at Umandawa. Umandawa frequently inspires these progressive ideals, yet there are contradictions owing to its use of borrowed ideas from diverse sources. That is, Umandawa sustains itself by modernising its understanding of Buddhism that reflects contemporary ideas, and at the same time fulfils the traditional concerns of Buddhists in modern Sri Lanka.

**Funding:** This research received external funding.

**Institutional Review Board Statement:** The study was conducted in accordance with the Declaration of Helsinki, and approved by the Human Ethics Committee, Otago University, New Zealand (12/036 and 11 April 2019).

**Informed Consent Statement:** Informed consent was obtained from all subjects involved in the study.

**Data Availability Statement:** Data available on request due to restrictions, e.g., privacy or ethical. The data presented in this study are available on request from the corresponding author. The data are not publicly available due to [research ethics].

**Conflicts of Interest:** The author declares no conflict of interest.

## Notes

[1]    I investigate one of Ven Siri Samanthabadra's recent projects, Umandawa, to learn about the Buddhist reforms he has envisioned. The fieldwork for this research was carried out in October 2019, along with interviews of *āśramavāsī* (residing) *bhikkhu*-s, *bhikkhunī*-s, and lay devotees who visited the Umandawa monastery.

[2]    In my research, it was challenging to get *bhikkhunī*-s to participate in the study as they were very reluctant and not ready to speak. I have had to ask permission from *āśramavāsī bhikkhu*-s to interview *āśramavāsī bhikkhunī*-s, who were relatively shy compared to *bhikkhu*-s.

[3]    Siyam *nikāya* adopted a new policy of only bestowing higher ordination on members of highest caste (Goyigama) in Kandyan kingdom (1747–1782), see in Ananda (Abeysekara 1999, p. 257).

[4]    When the Siri Dhamma's disciple monks applied for the higher ordination from the Malwatta temple, Kandy, only fourteen were selected out of twenty-four and other applications were rejected based on the lower castes they belong to. The rest of the monks had to obtain their higher ordination from other monastic fraternities in the island.

[5]    Current Sri Lankan *bhikkhunī* communities in many ways group around monks such as in the cases of Dambulla, Naugala, Dekanduwala, etc.

[6]    Uposatha refers to the Buddhist monastic assembly's fortnightly gatherings at the full moon and new moon to reaffirm the precepts of discipline.

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
