# Peer review of "Umandawa: Buddhist Transformation in Modern Sri Lanka"

_religions, doi:10.3390/rel14010118_

Round 1

Reviewer 1 Report

The topic is interesting and the author has certainly put a lot of effort and energy into gathering accurate and meaningful information. However, it seems to me that he / she began a discourse on the religious level and then developed it from an anthropological point of view. The article lacks the synthesis of these two aspects. For publication I would recommend curating this merger. Otherwise the result may not be clear.

Author Response

Point 1:

I believe that extensive English editing, which has already been completed, would solve the problem of synthesis of the two aspects (religious and anthropological). Please see the attachment.

Reviewer 2 Report

A very interesting and carefully argued article.  I am more a specialist in Indian/South Asian Sanskrit philosophy, than issues of modernity is Sri Lanka.  I recommend that this essay be reviewed by a specialist in Sri Lankan Buddhism.

Author Response

Point 1:

Please see the attachment for the English editing.

Reviewer 3 Report

Interesting manuscript based on fieldwork particularly about Buddhist nuns and female devotees in contemporary Sri Lanka but the article lacks academic rigor and could benefit from professional English editing.

General comment: For someone not familiar with Sri Lankan history it is hard to follow the first couple of pages. The author could perhaps add a paragraph at the beginning in which he/she outlines the history of Sri Lankan Buddhism in the 20th century to provide some historical context.

Line comments:

25: Heinz Bechert is an extremely outdated source. Any critical comments on his interpretation of Buddhism? You may want to check out Donald Lopez’s work on the “Scientific Buddha.”

73: could you explain more closely what a “traditional Sinhalese Buddhist ethos” is?

80: which ideological shift are you referring to? Please clarify.

175: “fieldwork, 2019” is an imprecise form of reference

328: what is the Siyam nikaya and why is it important to mention here?

687: “spiritual tourism”: an interesting term; I wonder in which other contexts it has been used?

709: did “canonical Buddhism” ever exist as such?

Author Response

Point 1: Extensive English editing was done (please see the attachment).

Point 2: Additional reading suggestions were included that cover the history of Sri Lankan Buddhism in the 20th century.

Line comments:
25: - Additional references were included to plug the shortfall in Heinz Bechert's work.
73: With reference, examples of traditional Sinhalese Buddhist ethos were incorporated.
80: Changed into “social, political and cultural changes in modern Sri Lanka”
175: Replaced with (fieldnotes, 19, October 2019)
328: Siyam nikāya as the leading fraternity in the country.
687: “spiritual tourism” - Spiritual tourists also visit Sri Lanka on the purpose of pilgrimage, cultural heritage, meditation and yoga etc.

709: Delete the term canonical Buddhism.

Round 2

Reviewer 1 Report

Conclusions in an article are very important. They have to pull the strings of the main points of the writing. I recommend developing the conclusion paragraph following these guidelines, which means expanding on the points mentioned.

Author Response

I carefully reviewed the article's citing and eliminated those that were superfluous. I also finalized the reference list by including any missing citations. I think the conclusion already summarizes and addresses the main points of what is presented in the abstract and the text, but with some alterations.

Reviewer 3 Report

the author fixed the major issues with this article

Author Response

Edited version is attached herewith.
